# Update on Extracellular Vesicle-Based Vaccines and Therapeutics to Combat COVID-19

**DOI:** 10.3390/ijms231911247

**Published:** 2022-09-24

**Authors:** Tamanna Mustajab, Moriasi Sheba Kwamboka, Da Ae Choi, Dae Wook Kang, Junho Kim, Kyu Ri Han, Yujin Han, Sorim Lee, Dajung Song, Yong-Joon Chwae

**Affiliations:** 1Department of Microbiology, School of Medicine, Ajou University, Suwon 16499, Korea; 2Department of Biomedical Science, Graduate School of Ajou University, Suwon 16499, Korea

**Keywords:** extracellular vesicles, COVID-19, SARS-CoV-2, exosomes, therapeutics, vaccine

## Abstract

The COVID-19 pandemic has had a deep impact on people worldwide since late 2019 when SARS-CoV-2 was first identified in Wuhan, China. In addition to its effect on public health, it has affected humans in various aspects of life, including social, economic, cultural, and political. It is also true that researchers have made vigorous efforts to overcome COVID-19 throughout the world, but they still have a long way to go. Accordingly, innumerable therapeutics and vaccine candidates have been studied for their efficacies and have been tried clinically in a very short span of time. For example, the versatility of extracellular vesicles, which are membrane-bound particles released from all types of cells, have recently been highlighted in terms of their effectiveness, biocompatibility, and safety in the fight against COVID-19. Thus, here, we tried to explain the use of extracellular vesicles as therapeutics and for the development of vaccines against COVID-19. Along with the mechanisms and a comprehensive background of their application in trapping the coronavirus or controlling the cytokine storm, we also discuss the obstacles to the clinical use of extracellular vesicles and how these could be resolved in the future.

## 1. Introduction

Coronavirus disease 2019 (COVID-19), caused by severe acute respiratory syndrome coronavirus-2 (SARS-CoV-2), a single-stranded, positive-sense RNA virus, was first identified in Wuhan, China, in 2019 and was declared a pandemic by the World Health Organization, with over 572 million confirmed cases and 6.3 million deaths by 25 July 2022 [1]. SARS-CoV-2 enters the host cell through its spike protein by binding to the receptor binding domain (RBD) of the S1 subunit to the angiotensin-converting enzyme 2 (ACE2), a receptor of the host cell surface, thereby initiating infection; this is further accompanied by the S2 subunit with the help of cellular proteases, fusing the virus the cell and successfully releasing the viral genome into the cell [2]. The spike protein of SARS-CoV-2 is the most important target for researchers in the development of therapeutics and vaccines for blocking infection compared with other interacting mediators [3,4]. Symptoms of COVID-19 can be mild, for example, rhinitis, fever, cough, sore throat, and shortness of breath, or they can progress to severe symptoms such as acute respiratory distress syndrome (ARDS), pneumonia, and acute lung injury, where mechanical ventilation is needed [5]. So far, researchers have focused on not only the pathogenicity of the virus itself but also the counter-immune response to it. Extracellular vesicles (EVs) are now considered one of the important entities in fighting against COVID-19 due to their roles in SARS-CoV-2 pathogenesis and their usefulness as novel therapeutics, delivery vehicles for drugs, and vaccine platforms [6,7,8]. 

EVs, comprising a variety of nano-scale vesicles ranging from 50 to 1000 nm in size, are released from all types of cells carrying a variety of lipids, proteins, and nucleic acids in a more protective manner than un-enveloped circulating biomolecules such as antibodies and cytokines from cellular DNases, RNases, proteases, and other degrading materials, due to the presence of the lipid bilayer membrane. Moreover, they have been shown to be nontoxic and unable to induce immune response, depending on the cell type [9], which is why they have been engineered to express targeting receptors or ligands artificially [10] or as a drug delivery system in cancer [11] and other various inflammatory diseases [12] and viral infections [13]. Similarly, EVs have recently been utilized in numerous ways as novel strategies against COVID-19, including for diagnostic purposes, therapeutics, and vaccine development [14].

In this review, we aim to summarize currently developed or developing EV-based vaccines and therapeutics produced by manipulating extracellular vesicles against COVID-19 to date, as summarized in Table 1. This includes the engineering of EVs, expressing a part of the spike (S), RBD of SARS-CoV-2; full-length S or ACE2 of the host cell for trapping virus; or reducing the cytokine storm depicted in Figure 1. Finally, we will highlight the issues and associated with developing EVs against COVID-19.

### Extracellular Vesicles

Extracellular vesicles, which are surrounded by a membranous lipid bilayer, are released into the extracellular space by all types of cells for the transport of cellular cargo to regulate a variety of biological processes, subsequently playing an essential role in intercellular communication [32,33].

EVs can be generated through one of two pathways: the endosomal pathway, which consists of the internalization of membrane proteins through the endocytic pathway being sequentially maturated into early endosomes, late endosomes, and multivesicular bodies, and the non-endosomal pathway, which refers to the externalization of membrane enclosing internal cellular cargos and membrane proteins in the way to first form buddings and then pinch off to form vesicles [34]. Even though EVs are secreted by both prokaryotic and eukaryotic cells, there has been little research so far on the secretion from prokaryotes; however, extensive studies have been performed on the release of EVs from mammalian cells such as neuronal cells [35], endothelial cells [36], mesenchymal stem cells (MSCs) [37], and epithelial cells [38]. They are found in many biological fluids, i.e., blood [39], synovial fluid [40], milk [41], urine [42], and saliva [43]. Although during the process of synthesis and secretion they share similar markers that make their classification difficult, on the basis of size, degree of similarity, and biogenetic mechanism, there are three major types of extracellular vesicles, i.e., exosomes, microvesicles, and apoptotic bodies [34]. However, the term EVs is used for exosomes and microvesicles in this review, as recommended by the International Society of Extracellular Vesicles (ISEV) guidelines, because of the heterogeneous nature of their preparation [44], unless otherwise specified in the study

Exosomes are characterized as the smallest EVs, ranging from 40 to 150 nm in diameter and originating from the endocytosis of the cellular membrane, resulting in the formation of multivesicular bodies (MVBs) by the inward invagination of the late endosomal membrane with the role of endosomal sorting complex required for transport (ESCRT) proteins [45]. Currently, exosomes can be purified using several methods, such as differential ultracentrifugation, rate-zonal centrifugation, ultrafiltration, poly-ethylene-glycol (PEG)-based precipitation, immunoaffinity capture, microfluidics, and size-exclusion capture [46]. Most popularly, exosomes are isolated through ultracentrifugation at 100,000× *g* after apoptotic bodies and microvesicles are removed, which are comparatively larger in size, and if further purification is needed, rate-zonal centrifugation is performed using the Optiprep gradient [47]. Exosomes express specific proteins, for example, Alix, TSG101, and clathrin, involved in the endocytosis of the plasma membrane and MVB formation. Similarly they express RAB proteins and annexins involved in membrane trafficking, and also tetraspanins, i.e., CD63, CD9, and CD81. The enrichment of these proteins make exosomes easily distinguishable for researchers from other non-classical exosomes that do not express exosomal markers such as CD63, CD9, and CD81 [48].

Microvesicles are defined as large vesicles [44] formed by the outward protrusion of the plasma membrane and range from 100 to 1000 nm in size but on average are 250–400 nm in diameter [49,50]. The pinching off and detachment of the membrane occurs at specific places and is influenced by the distribution of phospholipids along with the phosphorylation of myosin mediated by Rho-kinase along with contractile machinery [51,52], where they enclose nearby biomolecules, cell surface proteins, and fragmented rRNA and mRNA, which are planned to be trafficked towards the plasma membrane [53,54]. Microvesicles are composed of lipids displaying plasma membrane receptors and molecules of the cells from which they originated. They are smaller than apoptotic bodies and can be isolated from biofluids and cellular supernatants by centrifugation at 10,000× *g* after removing cell debris containing apoptotic bodies [34].

Apoptotic bodies, the largest among EVs, range from 1 to 5 um in diameter and are generated during the programmed self-destruction of the cell, and they can be pelleted down at speeds of 2000–4500 g; they share markers with microvesicles and exosomes and hence are distinguished from other vesicles based on size [44]. They originate from the protrusion and blebbing of the membranes of dying cells that contain intact cellular organelles and genomic DNAs, damaged nucleic acids, and randomly packaged cargo [55]. Cell death can be induced during normal physiological pathways or through any pathological pathways that initiate from the blebbing of the membrane, resulting in the formation of apoptotic projections such as microtubular spikes [56], apoptopodia [57], and beaded apoptopodia [55]. Apoptotic cells release “find-me” signals to recruit phagocytes and “eat-me” signals such as phosphatidylserine (PS) and exposed outside apoptotic bodies that bind to receptors in phagocytic cells, mediating the engulfment of apoptotic bodies and finally resulting in the clearance of apoptotic cells by phagocytes [58,59,60].

## 2. Roles of Extracellular Vesicles in COVID-19 Pathogenesis

### 2.1. Pathogenesis of COVID-19

The disease progression of COVID-19 is represented by three phases, depending on the clinical status: the viremia phase, the acute phase (pneumonia phase), and the recovery phase [61,62]. During the viremia phase, early infection by the virus is initiated in the epithelial cells of the nasal cavity and the larynx, the major site of virus entry, through angiotensin-converting enzyme 2(ACE2) receptors with the help of transmembrane serine protease 2 (TMPRSS2) [63,64]. 

Mature SARS-CoV-2 enters the alveolar epithelial cells after replicating itself in upper respiratory tracts using host cell machinery such as RNA polymerase, ribosomes, and cellular enzymes to synthesize its structural proteins such as spike, envelope, membrane, and nucleocapsid (S, E, M, and N) and other accessory proteins [2]. The virus divides itself into high numbers and enters the bloodstream from the lungs, resulting in viremia, which again attacks various organs including the lungs, kidneys, and gastrointestinal tracts [65]. In the acute phase (pneumonia phase), pattern-recognition receptors (PRRs) present in the immune cells are attracted to lung tissues, recognize danger-associated molecular patterns (DAMPs) from the destructed tissues and pathogen-associated molecular patterns (PAMPs) from the virus, and produce a variety of pro-inflammatory mediators such as cytokines, chemokines, and inflammatory mediators that are necessary for adaptive immune responses against SARS-CoV-2. In addition, in lymphopenia, a significant reduction in the number of lymphocytes is observed in patients in this phase [66]. 

Most patients progress to a convalescent phase by acquiring adaptive immunity to SARS-CoV-2, along with a decline in virus titers and a recovery of blood lymphocytes [67]. A small portion of patients from the pneumonia phase develop a severe phase. Most patients who progress into the severe phase are either immune-compromised or old in age and suffering from underlying chronic diseases such as diabetes, hypertension, respiratory diseases, and cancer [68]. In the severe phase, lymphopenia is worsened, and the increase in serum pro-inflammatory mediators is sustained [69]. An increase in cytokines and chemokines attracts other inflammatory cells such as neutrophils and monocytes to the lung tissues, leading to an influx in inflammatory cells that results in an increase in pathogenic inflammatory cells [70], finally causing a cytokine storm [71,72]. Pro-inflammatory cytokines such as tumor necrosis factor-alpha (TNF-α) and interleukins 1 and 6 (IL-1 and IL-6) increase permeability in the host’s vascular system by the dilation of smooth muscle and the contraction of the endothelial cells of blood vessels, which leads to alveolar edema and alveolar collapse, thereby leading to refractory hypoxemia and finally ARDS. Thus, the exchange of gases in the lungs is impaired, thereby increasing respiratory and heart rate to compensate for the oxygen deficiency, which makes breathing extremely difficult for the patient [71,72,73]. At the same time, an excess of circulating cytokines causes other tissues to induce a systemic inflammatory response. Moreover, blood pressure is significantly reduced due to the vasodilation of the organs and tissues, resulting in multisystem organ failure [71]. In addition to ARDS, disseminated intravascular coagulation is a common critical factor in multisystem organ failure [74].

### 2.2. EVs in COVID-19 Pathogenesis

It is highly interesting to consider the developing theory that EVs contribute to the dissemination and persistence of genetic material and proteins of SARS-CoV-2 [75] due to the similarity in the entrance, budding, and mechanisms of biogenesis during infection. Previous research has already shown that EVs enhance infections caused by CMV, HIV-1, or HSV-1 by transferring viral proteins and genetic material from infected cells to healthy cells [76,77,78]. Tetraspanins such as CD9 are one of the most widely expressed proteins on the surface of EVs and are thought to work together with TMPRSS2 to cleave fused viral glycoproteins and speed up the entrance of Middle East respiratory syndrome coronavirus (MERS-CoV) into lung cells [79]. Additionally, CD9 enhances lentiviral infection and improves transduction efficiency in immune-competent cells such as T lymphocytes and B cells [80]. In SARS-CoV-2 infection, EVs expressing ACE2 on their surface are suggested to be responsible for spreading and accelerating infection by assisting viral entry into cells [81]. EVs from the plasma of COVID-19 patients incorporate SARS-CoV-2 spike-derived fragments of proteins and RNA, albeit in a low copy number, but are able to progress the disease and induce immune responses [6,82,83]. In other respects, ACE2 on EVs prevents SARS-CoV-2 infection through its function as a decoy receptor [16,17]. Given that EVs have some common physical and biogenic characteristics with viruses, EVs derived from convalescent patient serum also express spike protein on their surface and act as a decoy target for neutralizing antibodies by their competitive inhibition of binding between the antibodies and the mature virion [82].

## 3. EV-Based Therapeutics against SARS-CoV-2

The method of communication of EVs from cell to cell is by transferring intraluminal EV cargo such as proteins, lipid, mRNA, and miRNA, under normal physical conditions and also in diseased states, which makes EVs very interesting. EVs have hence been exploited by researchers as a therapeutic modality for delivering substances of interest (15). EVs on one hand have been engineered to evoke the immune response in terms of immunotherapies [84], while on other hand, they have been utilized as delivery drugs and immune modulators to treat various human diseases such as solid and hematologic cancers, autoimmune diseases, and infectious diseases [85,86]; now EVs have been used as novel therapeutics against COVID-19, using multiple approaches to blocking infection to healthy cells [16,87].

### 3.1. Extracellular Vesicles Tagged with RBD

The very first target of the SARS-CoV-2 virus is type 2 alveolar epithelial cells of the lungs, where it binds to the highly expressing ACE2 receptors [88]. Although EVs have a great capacity as a natural carrier to deliver cargos to the recipient cells, a majority of externally injected EVs are absorbed by the liver, spleen, and pancreas instead of being taken up by the specific target. Therefore, for the satisfactory localization of specifically targeted therapeutic particles, it is suitable to tag EVs with tissue-specific peptides or antibodies targeting specific antigens [89]. 

In previous studies, vesicular stomatitis virus-G protein (VSVG) has been used as a fusion backbone with a variety of reporter proteins that include luciferase, green fluorescent protein (GFP), and red fluorescent protein (RFP) for the tracking and detection of EVs [90]. Recently, VSVG has been engineered to be fused with RBD of SARS-CoV-2 virus protein, the key domain in virus attachment and entry, by replacing the ectodomain of VSVG with RBD, resulting in the production of pseudoviral particles expressing RBD–VSVG fusion protein, without changing the physical properties of the modified EVs [15]. This is because it has been found that EVs and VSVs have similar lipid envelope compositions resulting from shared intracellular trafficking [91].

The association between the RBD of SARS-CoV-2 and the ACE2 of the host has been thought to be the most crucial to be targeted for vaccine development [92], antibody neutralization, and small-molecule inhibitors [93], which block the entry of virus into the cell. The host cell receptor ACE2 is present in most cell types, including the heart, intestine, kidneys, and lungs, among which the lungs show the highest expression of ACE2s and are proven to have a stronger affinity for SARS-CoV-2 than other organs [88,94,95,96]. Similarly, it has been found that the cellular presence of ACE2 receptors was needed for the entry of RBD-tagged EVs, which is why they were successfully targeted to the lungs and other tissues expressing ACE2 and reduced the infection of SARS-CoV-2 by delivering siRNA incorporated into RBD-tagged EVs, which was conducted in a transgenic hACE2 mouse model. This study suggested that SARS-CoV-2 infection could be efficiently treated by the delivery of antiviral drugs and that RBD-tagged EVs could be a potential therapeutic approach for other diseases [15].

### 3.2. EVs Expressing Tetraspanins Fusion

Tetraspanins, CD9, CD63, CD81, CD82, and CD151, are transmembrane proteins concentrated in the plasma membrane lipid microdomain and largely expressed in the exosomes, which makes them exosomal markers. They have also been known to contribute to the biogenesis and cargo sorting of exosomes [97]. In the field of COVID-19 therapeutics, engineered EVs expressing a novel fusion of a tetraspanin, i.e., CD63 with anti-SARS-CoV-2 nanobody, inhibited the binding of SARS-CoV-2 to ACE2, thereby neutralizing the SARS-CoV-2 infection [4]. Similarly, EVs expressing soluble ACE2 on their surface by the fusion of truncated scaffold of CD9 serve as decoy receptors for SARS-CoV-2 and block infection by SARS-CoV-2 variants including D614G, δ, and β, as well as the wild type [3]. Consistent with the current data, EVs expressing ACE2 inhibited the infection of pseudotyped lentiviruses from variants of concern of SARS-CoV-2, which was even more enhanced along with TMPRSS2 [16].

In addition to engineered EVs expressing specific proteins for hindering the cellular entrance of SARS-CoV-2, EVs produced from cell lines such as Vero CCL-81 and Vero E6 infected with SARS-CoV-2 can display surface proteins of SARS-CoV-2 that can recognize alveolar macrophages. Therefore, these EVs can be used as vehicles encapsulating drug delivery platforms [98,99].

### 3.3. ACE2 Loading onto EVs to Block Virus Entry

Because of the quickly evolving variants of SARS-CoV-2, COVID-19 remains a matter of concern and a major challenge despite tremendous advances in vaccine development and novel therapeutics [100,101,102]. Therefore, it is crucial to develop a strategy to cope with newly emerging variants of SARS-CoV-2. In this critical situation, EVs expressing ACE2 (evACE2) could be an alternative strategy. Supporting these data, small EVs, as well as non-membranous extracellular nanoparticles, exomeres, are reported to express ACE2 on their surface which subsequently inhibit SARS-CoV-2 infection, acting as a decoy by binding to the virus [103]. 

Moreover, the loading of ACE2 onto EVs is highly dependent on ACE2 palmitoylation at two major sites, i.e., Cys141 and Cys498, called S-palmitoylation. EVs’ secretion of ACE2 from the plasma membrane is increased by palmitoylation through the zinc finger DHHC-Type palmitoyltransferase 3 (ZDHHC3) or decreased by de-palmitoylation through acyl protein thioesterase 1 (LYPLA1). During viral infection, ACE2 present on the membrane is secreted along with EVs, and it blocks infection with SARS-CoV-2 by binding to the RBD domain, preventing the binding of SARS-CoV-2 with cellular ACE2. Therefore, engineered EVs enriched with ACE2 on their surface by palmitoylation were bound efficiently to RBDs and subsequently neutralized both pseudo-typed and wild-type SARS-CoV-2 in human ACE2 transgenic mice [18]. 

### 3.4. CD24-Loaded EVs

CD24, expressed in a variety of cells, is a glycosylphosphatidylinositol (GPI)-anchored glycoprotein and is localized in lipid microdomain [104,105]; its interaction with Siglecs, which are mostly expressed by cells of the immune system, has been shown to transduce inhibitory signals on Siglec-expressing inflammatory cells [106,107,108]. Therefore, CD24 fused with Fc fragment of Ig (CD24-Fc) was tested to evaluate its effect in reducing over-activated inflammation in COVID-19, which is similar to human immunodeficiency virus type-1/simian immunodeficiency virus infection, where it provides protection to Chinese rhesus macaques (ChRMs) against disease progression [109,110]. Intriguingly, CD24 is also highly expressed in EVs, possibly due to its localization in lipid microdomain and GPI anchorage, prompting the development of CD24-expressing EVs as a novel therapeutic. Consequently, exosomes derived from CD24-overexpressing 293T cells, T-Rex™ (EXO-CD24), are developed for the treatment of COVID-19 cytokine storms with the format of inhalation. The rationale of this new drug is that exosomes expressing CD24 can attenuate the cytokine storm via activating anti-inflammatory immune cells through CD24 signaling by over-activating innate immune responses of SARS-CoV-2 patients [19].

### 3.5. EVs from Convalescents

Exosomes from the plasma of the convalescent phase of COVID-19 patients were recently reported to be equipped with all the components, including viral proteins, peptides, and RNAs for successful adaptive immune responses to SARS-CoV-2 [83]. 

The release of EVs expressing ACE2 (evACE2) was markedly increased in the convalescent serums of severely infected COVID-19 patients. They inhibited infections of SARS-CoV-2 including α, β, and δ strains with 135 times higher efficiency than recombinant human ACE2 (rhACE2), by competing with cellular ACE2 in the binding with RBD of SARS-CoV-2. In vivo, the ACE2-transgenic mouse model evACE2 was 60 to 80 times more potent than rhACE in inhibiting infection from both original and pseudotyped viruses and also protected mice from SARS-CoV-2-induced lung injury [17].

Following on this finding, virus-specific T cells should have similar capabilities with the plasma exosomes, and they are currently in clinical trials. COVID-19-specific T-cell-derived exosomes (CSTC-Exo) were purified from T cells of convalescent COVID-19 patients and expanded in vitro by stimulation with SARS-CoV-2-specific peptides and cytokines. An inhalable format of CSTC-Exo is now in clinical trials in COVID-19 patients with early pneumonia. Action mechanisms of CSTC-Exo have suggested that cytokines, including IFN-γ within the exosomes prepared from activated SARS-CoV-2-specific T cells, have anti-viral effects on COVID-19 patients [20]. 

Convalescent plasma infusion has been successfully tried in severe COVID-19 patients for treatment [111,112] and is now considered a possible treatment choice in severe COVID-19 patients not responding to existing therapies [113]. Among the components of convalescent plasma, it has been proposed that EVs from platelets are the major constituents responsible for the regeneration of damaged tissues [114]. Therefore, platelet-derived EVs could be used as a COVID-19 therapeutic. Supporting this notion, platelet-derived exosomes loaded with an anti-inflammatory drug, [5-(p-fluorophenyl)-2-ureido] thiophene-3-carboxamide (TPCA-1), have been shown to alleviate cytokine storm in pneumonia in a mice model [21].

## 4. EV-Based Vaccines against SARS-CoV-2

EVs have also been used as a novel platform of vaccines by manipulating them in numerous interesting ways, although lipid nanoparticle-based vaccines have successfully been used as delivery vehicles for SARS-CoV-2 mRNA. The lipid nanoparticle-based vaccine reportedly has some major shortcomings such as temperature instability, thus requiring the maintenance of ultra-low temperatures, as well as side effects and adverse reactions, and poor mucosal immunity [115,116,117]. Thus, EV-based vaccines can be proposed to overcome the drawbacks of lipid nanoparticle-based mRNA vaccines. Here, we want to introduce the achievements with significant advances among various projects recently forwarded.

Exosomes loaded with an mRNA-encoding spike and nucleocapsid proteins of SARS-CoV-2 induced long-lasting cellular and humoral immune responses in a mouse model and proved to be safer than other lipid nanoparticle-based mRNA vaccines [27].

Moreover, EVs purified from cells transfected with a SARS-CoV-2 spike DNA (CoVEXax™, Ciloa SAS) triggered antibody- and cell-mediated immunities in mice without using any adjuvants [28].

A candidate vaccine that was based on the lyophilized exosomes derived from lungs (S-Exos) expressing mRNA of SARS-CoV-2 spike protein [29] and exosomes conjugated with recombinant SARS-CoV-2 RBD (RBD-Exo) [30] was more potent in stimulating IgG and secretory IgA (SIgA) responses when delivered to rodents and nonhuman primates compared with synthetic nanoparticle liposomes loaded with the spike mRNA of SARS-CoV-2. These EV-based vaccines were stable, even at room temperature, for more than a month when formulated as either inhalable dry powder or in lyophilized form. They could also be directly distributed to the bronchioles and parenchyma of the lungs [29,30].

An intranasal vaccine based on bacterial EVs and outer membrane vesicles (OMV) of *Salmonella typhimurium* displaying RBD of SARS-CoV-2 produced high titers of neutralizing antibodies against Wuhan-type SARS-CoV-2, as well as delta variants, and protected immunized golden Syrian hamster from infection with SARS-CoV-2 with no loss of body mass and fewer viral titers in bronchoalveolar lavage fluids compared with the control [31]. Hence, it is believed that EV-based vaccines are ready to circumvent the current limitations of lipid nanoparticle-based mRNA vaccines.

## 5. Mesenchymal Stem Cell (MSC)-Derived EVs in COVID-19

MSCs are adult stem cells derived from multiple tissues such as umbilical cord, bone marrow, adipose tissue, and amniotic fluid and have the ability to regenerate and differentiate into cartilage, bone, nerve, muscle, and skin cells [118]. They are considered to be the most useful tool in regenerative medicine as they can repair damaged tissue and organs and release a variety of cytokines, chemokines, growth factors, and also EVs [119]. Compared with other cell types, MSCs are known to release tremendously high amounts of EVs, including exosomes. EVs from MSCs have been reported to have similar therapeutic effects to MSCs because they have active biomolecules analogous to MSCs and because EVs from MSCs are responsible for the paracrine effect of MSCs in MSC cell therapies [120]. Researchers have been testing stem-cell-derived EVs for the management of COVID-19, and the majority of these EVs are in clinical trials as summarized by Krishan et al. [121].

One of the therapeutically active parts of MSC exosomes is miRNAs inside exosomes. Thus, MSC exosomes containing miRNAs have been used as cell-free therapeutics in multiple diseases, such that miRNA in exosomes from umbilical cord-derived MSCs (UC-MSCs-Exo) inhibited hepatitis C virus (HCV) infection by silencing viral RNA in combination with other FDA-approved drugs [122]. Similarly, COVID-19 patients with mild pneumonia showed improved clinical outcomes when EVs were directly targeted to the lungs via inhalation by the nebulization of UC-MSCs-Exo and human adipose-derived exosomes (haMSC-Exos) without causing any side effects [22,23]. Owing to the regenerative and repair nature of MSC EVs as proved through in silico studies, miRNA from MSC EVs attenuated the cytokine storm, protected cells from damage, and blocked the activation of the coagulation pathway caused by COVID-19. Hence, these results show the effects of MSC EVs on multiple targets in COVID-19 [123]. 

Another promising therapeutic candidate based on exosomes derived from MSCs of allogeneic bone marrow (ExoFlo™, Direct Biologics, Austin, TX, USA) is currently being studied in human clinical trials. As a result, it has been proposed that it is very effective in moderate to severe COVID-19 patients. Given that MSCs-derived EVs are safer compared to MSCs because of their non-replicative, non-tumorigenic, and non-immunogenic nature, ExoFlo™ was quite useful in restoring plasma oxygen contents, lowering cytokine storm, and reconstituting immunity in infected patients during human trials [24].

MSC-Exo in COVID-19 patients may induce the differentiation of M2 macrophages by promoting the secretion of PEG2 and accordingly can inhibit inflammation by blocking the release of pro-inflammatory cytokines and can reduce the effects of pro-inflammatory cytokines through anti-inflammatory cytokines such as IL10. Similarly, MSC-Exo can regenerate and repair the damaged tissues through the induction of growth factors such as vascular endothelial growth factor (VEGF), keratinocyte growth factor (KGF), and hepatocyte growth factor (HGF), as already reported in multiple disease models [124,125,126]. Moreover, EVs derived from Wharton’s jelly MSC could be very effective in reducing nuclear the NF-kB-mediated cytokine storm in COVID-19 patients with diabetes mellitus or renal disease [26].

Other studies about EVs derived from amniotic fluid named Zofin™ have proved them to be successful in improving the conditions of three critical patients suffering from COVID-19 without showing any adverse effects or safety concerns [25,127]. 

## 6. Conclusions

In this review, we aimed to describe currently developing therapeutics and vaccine candidates based on EVs for COVID-19, together with their action mechanisms and comprehensive backgrounds for their application, as summarized in Table 1 and Figure 1. EVs have been an ideal natural therapeutic platform serving as a decoy or as a delivery vehicle carrying drugs and other cargo and thus communicating within or between cells in all organisms, from bacteria to humans. 

However, it is also true that there are several clear issues that need to be overcome for the clinical use of EVs to be possible. Most researchers in the field of EVs are pointing out common problems in the commercialization of EVs: extremely low yields of EVs from the large-scale production of conditioned media of cultured cells; isolation of EVs and the challenges in each method of EV isolation; discrepancies between produced batches of EVs from different cell types; maintenance of efficacy during storage, transportation, and biodistribution of EVs, including targeting EVs to the desired tissues or organs without loss of efficacy; and also the issue of long-term endurance by the human body during clinical studies despite having better safety compared to cell therapies. 

Considering these factors, we propose that most of the problems come from our incomplete understanding of the extreme heterogeneity of EVs. Our knowledge of the biogenesis of EVs in individual cells appears not to be sufficient despite significant advances in recent research. 

Moreover, specific cells release their unique EVs with peculiar cargos using different mechanisms of synthesis depending on the tissues of origin and the cell condition. We are aware that stressed cells can release different kinds of EVs from those of normal cells, depending on the kind of stress. 

Furthermore, the cell lines being used for the isolation of EVs for clinical purposes are limited to several non-cancerous cell lines and primary human cells because of their origin and biocompatibility issues. Therefore, a more precise understanding of the mechanisms of action of EVs, along with their biogenesis, is required for the commercialized usage of EVs to combat COVID-19 and other emerging diseases.

## Figures and Tables

**Figure 1 ijms-23-11247-f001:**
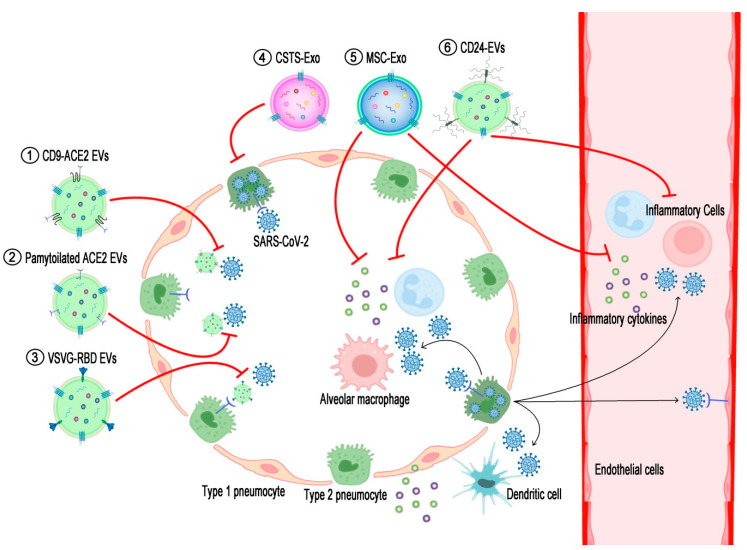
Schematic diagram displaying action mechanisms of EV-based therapeutics against COVID-19. SARS-CoV-2 enters and proliferates in the type 2 pneumocytes of the lung and then spreads into the interstitial tissue and bloodstream. Components of either viruses or dying cells activate alveolar macrophages and dendritic cells to recruit inflammatory cells to the lung tissue, resulting in the over-secretion of pro-inflammatory cytokines (cytokine storm). (1) EVs expressing ACE2-fused with CD9 (CD9-ACE2 EVs) or (2) palmitoylated ACE2 (palmitoylated ACE2 EVs) prohibit the binding of viruses to cellular ACE2. (3) EV-expressing RBDs of SARS-CoV-2 spike proteins fused with the stem region of the VSVG protein (VSVG-RBD EVs) target ACE2-expressing cells and thereby introduce anti-viral siRNAs to inhibit the proliferation of the viruses. (4) COVID-19-specific T-cell-derived exosomes (CSTS-Exo) show anti-viral effects on virus-infected cells by their cargo such as IFNγ. (5) EVs from mesenchymal stem cells (MSC-Exo) or (6) EVs expressing CD24 (CD24-EVs) can ameliorate the cytokine storm induced by over-activated inflammatory cells in the severe phase of COVID-19. The black lines briefly indicate the pathogenic pathways of SARS-CoV-2 infection in the airway, and the red lines denote the action mechanisms of the EV-based therapeutics interrupting the COVID-19 pathogenesis.

**Table 1 ijms-23-11247-t001:** EV-based therapeutics and vaccine candidates under development for COVID-19.

	EV Type/Origin	Purification	Mechanism of Action	Study	Format	
**Therapeutics**	EVs loaded with engineered VSVG to fuse RBD of SARS-CoV-2 virus including siRNA, HEK-293T cells (VSVG-RBD)	Filtration (0.22 mm), Ultracentrifugation	Trapping virus	In vitro/ in vivo	Injectable	[15]
	EVs displaying fusion of CD63 and anti-SARS-CoV-2 nanobody using HEK-293T cells (CD63-S)	Ultracentrifugation	Trapping virus	In vitro/ in vivo	Injectable	[4]
	EVs presenting fusion of truncated CD9 scaffold to display ACE2 using HEK-293T cells (CD9-ACE2)	Filtration (0.22 mm), ultrafiltration	Trapping virus	In vitro/ in vivo	Injectable	[3]
	Engineered EVs from 293FT cells expressing ACE2 and TMPRSS2	Size-exclusion chromatography	Trapping virus	In vitro/ In vivo	Injectable	[16]
	EVs isolated from severely infected COVID-19 patients serums that express ACE2	Ultracentrifugation (100,000× *g*)	Trapping virus	In vitro/ In vivo	Injectable	[17]
	EVs extracted from HEK-293T cells having a fusion of S-palmitoylated sequence with ACE2 (PM-ACE2-EVs)	Filtration (0.22 mm), Ultracentrifugation (100,000× *g*)	Trapping virus	In vitro/ In vivo	Injectable	[18]
	Exosomes isolated from CD24 expressing 293-TREx™ derived from HEK-293 cells (CD24-EXO-TREx™)	Filtration (0.22 mm), precipitation (ExoQuick-CG)	Attenuating cytokine storm	Clinical trial	Inhalable	[19]
	Exosomes derived from COVID-19-specific T cells of convalescent patients (CSTC-Exo)		Attenuating cytokine storm	Clinical trial	Inhalable	[20]
	Platelet-derived EVs from plasma of convalescent carrying TPCA-1	Ultracentrifugation	Attenuating cytokine storm	In vitro/ In vivo	Injectable	[21]
	Exosomes from umbilical-cord-derived mesenchymal stem cells (UC-MSCs-Exo)	Ultracentrifugation	Treating mild pneumonia	Clinical trial	Inhalable	[22]
	Exosomes from human-adipose-tissue-derived mesenchymal stem cells (haMSC-Exos)		Treating mild pneumonia	Clinical trial	Inhalable	[23]
	Exosomes from bone-marrow-derived mesenchymal stem cells (ExoFlo™)		Attenuating cytokine storm	Clinical trial	Injectable	[24]
	Exosomes from amniotic-fluid-derived mesenchymal stem cells (Zofin™)		Treating COVID-19 long haulers	Clinical trial	Injectable	[25]
	Exosomes from Wharton’s jelly–derived mesenchymal stem cells	Ultracentrifugation	Attenuating cytokine storm	In vitro/ In vivo	Injectable	[26]
**Vaccine candidate**	Exosomes isolated from 293F cell loaded with mRNA expressing the immunogenic form of Spike and nucleocapsid proteins	Filtration, Size-exclusion chromatography	Adaptive immunity	In vitro/ In vivo	Injectable	[27]
	EVs derived from HEK-293T cells expressing Spike protein (CoVEXax™)	Filtration (0.22 mm), ultrafiltration, Size-exclusion chromatography	Adaptive immunity	In vitro/ In vivo	Injectable	[28]
	Exosomes purified from lung spheroid cells (Lung-Exo) and loaded with spike protein mRNA	Ultrafiltration	Adaptive immunity	In vitro/ In vivo	Inhalable	[29]
	Exosomes purified from lung spheroid cells (Lung-Exo) and conjugated with the RBD of spike protein	Ultrafiltration	Adaptive immunity	In vitro/ In vivo	Inhalable	[30]
	Bacterial OMV conjugated with RBD of spike protein	Ultracentrifugation	Adaptive immunity	In vitro/ In vivo	Inhalable	[31]

## Data Availability

Not applicable.

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
