# Peer review of "Update on Extracellular Vesicle-Based Vaccines and Therapeutics to Combat COVID-19"

_ijms, 2022, doi:10.3390/ijms231911247_

Round 1

Reviewer 1 Report

Despite the extended research on extracellular vesicles as therapeutic agents, the application of EV as a vaccine is quite new. The review provides an interesting and up-to-date overview of the latest findings in this field.

The work is well written, the cited literature is appropriate, the order of paragraphs is correct, and the figures and tables are clear and exhaustive.

The English form is correct. Some sentences are very long (more than three lines), so I suggest shortening the sentences to simplify the comprehension for the readers, when applicable.

In paragraph 1.1, page 56, please correct 1000,000g with 100,000g.

Moreover, the classification of exosomes and microvesicles is formally correct, but a bit dated by now because it is difficult to distinguish the two classes in laboratory practice. For this reason, ISEV guidelines (MISEV2018: https://doi.org/10.1080/20013078.2018.1535750) recommend using the term “extracellular vesicles” to include both classes. So I suggest adding a few lines at the end of the paragraph to clarify this point.

In paragraph 3, page 9, there are a couple of numbers in round brackets. Please check if they are references and correct.

Reviewer 2 Report

In the manuscript by Mustajab and co-authors entitled “Current Update on Extracellular Vesicles-Based Vaccines and Therapeutics to Combat COVID-19” the authors reviewed the potential application of extracellular vesicles as a platform for vaccines or therapeutics to fight COVID-19 disease. The review brings interesting information and is useful for readers working in the area.

My major comment is that the English language needs revision.

The authors should Improve the figure legend, what are the red and black lines? Including numbers would facilitate the figure interpretation linked with the legend.
